# Modeling Gastrointestinal Tract Wet Pool Size in Small Ruminants

**DOI:** 10.3390/ani13182909

**Published:** 2023-09-13

**Authors:** Paola R. Ribeiro, Marcelo Gindri, Gilberto L. Macedo Junior, Caio J. L. Herbster, Elzania S. Pereira, Bruno Biagioli, Izabelle A. M. A. Teixeira

**Affiliations:** 1Department of Animal Science, São Paulo State University, Jaboticabal 14884900, SP, Brazil; paola.rezenderibeiro@gmail.com (P.R.R.); bruno.biagioli@unesp.br (B.B.); 2UMR Modélisation Systémique Appliquée aux Ruminants, AgroParisTech, INRAE, Université Paris-Saclay, 91120 Palaiseau, France; gindri.marcelo@gmail.com; 3Department of Animal Science, Federal University of Uberlândia, Uberlândia 38402018, MG, Brazil; gilberto.macedo@ufu.br; 4Department of Animal Science, Federal University of Ceará, Fortaleza 60356000, CE, Brazil; julioherbster@gmail.com (C.J.L.H.); elzania@hotmail.com (E.S.P.); 5Department of Animal Veterinary, and Food Sciences, University of Idaho, Twin Falls, ID 83301, USA

**Keywords:** diet, goats, gut fill, intake, intrinsic factors, meta-regression, neutral detergent fiber, prediction models, sheep, small ruminants

## Abstract

**Simple Summary:**

Accurately measuring the wet pool size of the gastrointestinal tract (GITwps) provides valuable information for understanding how digesta behaves in relation to the various factors that affect it. These factors include species anatomy, the animal’s physiological state, body weight, diet, and ingestion level. Access to this information allows for the development of improved nutritional strategies, the refinement of predictive models for passage rate, and enhanced predictions of dressing percentage and empty body weight. Moreover, the development of models for predicting GITwps is crucial because the current methods are laborious, time-consuming, expensive, and often require animal slaughter or involve the use of inaccurate markers. This study aimed to develop comprehensive models for predicting GITwps of small ruminants using a meta-regression approach. The prediction models confirm that the relationship between GITwps and BW is robust for animals fed a solid diet. Additionally, the physiological stage, such as pregnancy, influences GITwps, and the effect of neutral detergent fiber intake on GITwps is consistent across different species.

**Abstract:**

The gastrointestinal tract (GIT) wet pool size (GITwps) refers to the total amount of wet contents in GIT, which in small ruminants can reach up to 19% of their body weight (BW). This study aimed to develop models to comprehensively predict GITwps in small ruminants using a meta-regression approach. A dataset was created based on 21 studies, comprising 750 individual records of sheep and goats. Various predictor variables, including BW, sex, breed, species, intake level, physiological states, stages and types of pregnancy, dry matter intake, and neutral detergent fiber intake (NDFI), were initially analyzed through simple linear regression. Subsequently, the variables were fitted using natural logarithm transformations, considering the random effect of the study and residual error, employing a supervised forward selection procedure. Overall, no significant relationship between GITwps and BW (*p* = 0.326) was observed for animals fed a milk-based diet. However, a strong negative linear relationship (*p* < 0.001) was found for animals on a solid diet, with the level of restriction influencing GITwps only at the intercept. Furthermore, the prediction of GITwps was independent of sex and influenced by species in cases where individuals were fed ad libitum. Pregnant females showed a noticeable reduction in GITwps, which was more pronounced in cases of multiple pregnancies, regardless of species (*p* < 0.01). The composition of the diet was found to be the primary factor affecting the modulation of GITwps, with NDFI able to override the species effect (*p* < 0.0001). Overall, this study sheds light on the factors influencing GITwps in small ruminants, providing valuable insights into their digestive processes and nutritional requirements.

## 1. Introduction

The gastrointestinal tract (GIT) wet pool size (GITwps) is the total amount of wet contents present in the GIT [1]. In small ruminants, it can reach up to 19% of body weight (BW) [2]. The GITwps is influenced by various factors, including feed and water intake [3], diet composition [4,5], and physiological states such as animal growth [6] and pregnancy [7]. However, little is known about the effects of species, breed, and sex of small ruminants on GITwps. Understanding the dynamics of GITwps is crucial for developing effective feeding strategies, accurate prediction models for GITwps, refining predictive models for passage rate, and improving the prediction of dressing percentage and empty BW (EBW).

Developing models to accurately predict GITwps in animals is important because the current methods for measurement are laborious, time-consuming, expensive, and either require animal slaughter [8] or involve the use of inaccurate markers [6]. Measurement of GITwps is essential in studies that evaluate digestion and nutrient passage through the GIT [9,10,11]. These studies require the use of the content from each compartment of the tract as an input variable [12,13]. Therefore, accurate prediction of GITwps will facilitate the development of models to predict other digestive traits, such as digesta passage rate.

The GITwps also contributes to the variation in BW measurements and consequently affects the precision in the prediction of weight gain as well as empty body weight. Several models have been developed to predict EBW for cattle [14,15,16], sheep [2], and goats [17]. However, only a few studies have focused on predicting GITwps [18,19,20], and none of them have been conducted with small ruminants considering the relationship between BW, different species, sex, physiological state, and diet composition. Therefore, this study aims to develop models to comprehensively predict GITwps in small ruminants using a meta-regression approach.

## 2. Materials and Methods

### 2.1. Data Set

All procedures used in the studies were thoroughly reviewed and approved by the Animal Care and Use Committee of the respective university where the studies were conducted. A data set was compiled from twenty-one studies comprising 750 individual records (Table 1). Thirteen studies were conducted with goats of different sexes (intact males, n = 218; castrated males, n = 116, and females, n = 110), breeds (Saanen, n = 319; ½ Boer × ½ Saanen, n = 45; ¾ Boer × ¼ Saanen, n = 21; indigenous, n = 38; and Oberhasli, n = 21), types of diet (suckling, n = 36; and solid-fed, n = 408), physiologic states (pregnant adult, n = 42; and growing, n = 402), stages of pregnancy in days (50, n = 6; 80, n = 12; 110, n = 12; and 140, n = 12), and type of pregnancy (single, n = 18 and twin, n = 24) (Table 1). The remaining eight studies were conducted with sheep of different sexes (intact males, n = 161; castrated males, n = 26; females, n = 119; and), breeds (Santa Ines, n = 170; ½ Dorper × ½ Santa Ines, n = 54; and Morada Nova, n = 82), physiologic states (pregnant adult, n = 73; and growing, n = 233), stages of pregnancy in days (90, n = 14; 110, n = 15; 130, n = 16; and 140, n = 28), and the number of fetuses (1, n = 28; 2, n = 37; and 3, n = 8) (Table 1).

In general, the diets consisted of dehydrated maize (*Zea mays*, whole maize plants chopped when the kernel milk line was approximately two-thirds down the kernel and then dehydrated), Tifton hay (*Cynodon* spp.), Elephant grass hay (*Pennisetum purpureum*), or maize silage; maize grain, soybean (*Glycine max*) meal, soybean oil, limestone, wheat (*Triticum aestivum*) bran, ammonium chloride, dicalcium phosphate, sodium chloride, and premix. The roughage-to-concentrate ratio ranged from 25:75 to 89:11. Further details can be found in each original study (Table 1). For all studies, dry matter intake (DMI) and neutral detergent fiber intake (NDFI) were individually and daily recorded. Animals were weighed on the same day of slaughter, and this record of weight was used for the calculations of DMI (g/kg BW) and NDFI (g/kg BW). The NDFI expressed as diet concentration (NDFI, g/g DMI) was calculated by the ratio of NDF daily intake (g) and DM daily intake (g). The daily intake was the average daily intake registered during the last five days prior to slaughter. 

The studies conducted by Härter et al. [32] and Macedo Junior [36] involved pregnant adults slaughtered on specific days of pregnancy, with BW ranging from 29.14 kg to 66.80 kg. In the remaining studies, animals were slaughtered at a target BW ranging from 5 kg (5.39 ± 0.55 kg BW) to 50 kg (51.75 ± 8.32). The BW utilized in the equations was recorded as full weight. In all studies, the GIT was removed at slaughter, and its weight was recorded before and after emptying to determine the GIT wet pool size.

### 2.2. Models’ Development

In this study, a supervised forward selection procedure was employed to select the models for predicting GITwps in small ruminants. Predictor variables were added or removed from the set of explanatory variables based on the pre-specified *p*-value, considering their uniform distribution across the range of BW in the dataset, as well as the goodness of fit of the models (i.e., Akaike’s Information Criterion (AIC) with a small sample correction (AICc) and concordance correlation coefficient (CCC)). Predictor variables with *p*-values smaller or equal to 0.05, homogeneously distributed across the BW range in the dataset, and with the potential to enhance model fitness were retained and included in the final selected models. The following predictor variables were considered: BW (kg), intake level (i.e., ad libitum or restriction levels), sex, breed, species, DM intake level (g/kg BW), NDFI expressed either as diet concentration (NDFI, g/g DMI) or as BW relative intake (NDFI, g/kg BW), physiological state (i.e., pregnant adult and growing), stage of pregnancy in days, and type of pregnancy (i.e., single or twin). Additionally, to examine the non-linear relationship between GITwps and the candidate predictor variables, as well as to address residual heteroscedasticity and enhance the predictive ability of models, four linearly selected equations were fitted using the natural logarithm transformed GITwps and the predictor variable BW.

The data were fitted to the models using lme the function from the nlme package [42] in R (R Core Team), considering the random effect of the study and residual error. Significance was declared at *p* ≤ 0.05. The coefficient of determination (*R*^2^) was computed using the r.squaredGLMM function from the MuMIn package [43]. The variance inflation factor (VIF) was computed using the vif function from the car package [44] in R (R Core Team), and all the selected models showed a VIF smaller than 10.

### 2.3. Model’s Evaluation

The evaluation of the models consisted of two different steps [45]. In the first step, the predictor variables were ranked based on their magnitude in estimating the predicted variable. In the second step, the selected models’ predictive power was assessed using a 4-fold cross-evaluation repeated over a thousand iterations. 

The mean square prediction error (MSPE, defined as the averaged squared differences between observations and predictions from the fixed effect part of the model only) was used to decompose the uncertainty of prediction into errors associated with mean bias, systematic bias, and random errors. The positive square root of MSPE (RMSPE) was used as an indicator of the average uncertainty of prediction (accuracy). Lin’s CCC was also used as a measure of goodness of fit and agreement between observations and predictions [46]. The coefficient of determination (*R*^2^) was used as an indicator of the precision of predictions. 

## 3. Results

### 3.1. Models’ Development

The selection of models began by testing BW (kg) as a predictor variable for GITwps (g/kg BW) using all observations in the dataset (n = 750). When suckling and solid diet-fed animals were evaluated together, no relationship (*p* > 0.05) was observed between GITwps (g/kg BW) and BW (kg) (Figure 1; [47]). Consequently, the relationship between the GITwps (g/kg BW) and BW (kg) was modeled separately for suckling animals (n = 36) and solid diet-fed animals (n = 714), as shown in Equations (1) and (2). Suckling animals presented a unique condition in which the GIT wet pool size (g/kg BW) is not influenced by BW (*p* = 0.326; *R*^2^ = 0.45), Equation (1), and Figure 1.

(1)
GITwps_suckling animals_ (g⁄kg BW) = 102.90 (±26.33) + 3.92 (±3.93) × BW (kg)


On the other hand, individuals on a solid diet presented a strong, linear, negative relationship (*p* < 0.001; *R*^2^ = 0.42) between GITwps and BW, as depicted in Equation (2) and Figure 1.

(2)
GITwps_solid-fed animals_ (g⁄kg BW) = 255.26 (±8.85) − 1.83 (±0.23) × BW (kg)


For the solid-fed animals, the effect of intake level (i.e., ad libitum or restriction levels) was incorporated into the model that includes BW as the predictor variable (i.e., Equation (2)). Intake level only influenced the intercept (*p* < 0.001) of the relationship between GITwps and BW. Individuals subjected to restriction levels exhibited a GITwps that was 5.13 (g/kg BW) smaller than individuals fed ad libitum. Therefore, Equation (3) (*R*^2^ = 0.42) was proposed to predict GITwps for individuals fed a solid diet ad libitum, and Equation (4) (*R*^2^ = 0.42) for individuals fed a solid diet with restriction levels.

(3)
GITwps_fed ad libitum_ (g⁄kg BW) = 252.53 (±11.24) − 1.78 (±0.29) × BW (kg)


(4)
GITwps_restriction levels_ (g⁄kg BW) = 247.40 (±9.85) − 1.43 (±0.35) × BW (kg)


In Equation (2), the effects of sex, breed, and species were also examined. Sex had a significant influence on the intercept (*p* = 0.029) but did not impact the slope of the relationship between GITwps and BW (*p* = 0.27). Intact males exhibited the highest intercept, followed by castrated males and females (268.79, 242.60, and 236.079 g/kg BW, respectively). Breed and species appeared to affect both the intercept and slope of the relationship between GITwps and BW (*p* ≤ 0.022). However, the results regarding breed should be interpreted with caution because not all breeds were evenly distributed across the entire BW range (Figure 2). Consequently, we opted to proceed without considering the breed effect in the model. 

To examine the effect of physiological state, days of pregnancy, and types of pregnancy (i.e., single or twin) on the relationship between GITwps (g/kg BW) and BW (kg), we incorporated these effects into Equation (3). We found that days of pregnancy and types of pregnancy only affected the intercept of the relationship between GIT and BW (*p* < 0.01), irrespective of species. Specifically, we propose a reduction in the intercept of 0.708 for a single pregnancy and 0.909 for multiple pregnancies (i.e., two or more fetuses) after 50 days of pregnancy.

Considering the unique characteristic of our dataset, which consists of individual observations, animals fed restriction levels within the same study had identical ingested diet compositions and were pair-fed. As a result, Equation (3) was re-fit using data solely from individuals fed ad libitum, and the effects of sex, species, intake, and diet composition were incorporated into the re-fitted Equation (3). The sex effect on Equation (3) was no longer significant when it was re-fitted using individuals fed ad libitum only (*p* = 0.088). However, species influenced both the intercept and slope of the relationship between GITwps and BW in individuals fed ad libitum only (*p ≤* 0.0166; Table 2; Equation (5)). Additionally, the inclusion of DMI (g/kg BW) showed statistical significance for sheep and goats (*p* ≤ 0.0059; Table 2; Equation (6)). On the other hand, the effect of NDF intake (g/kg BW) on the GITwps was independent of species, and the GIT content increased by 2.84 g/kg BW with increasing NDFI (Table 2; Equation (7)). Moreover, when the concentration of NDFI in the diet (g/g DMI) was added to Equation (3), it eliminated the species effect, suggesting that its effect on the GITwps is similar for both sheep and goats (Table 2; Equation (8)). Among all the fitted equations, the four selected equations presented in Table 2 showed the potential to accurately and precisely predict GITwps in small ruminants.

Given that animal growth does not follow a linear pattern, nonlinear functions provide a more biologically meaningful interpretation that aligns with reality. Therefore, in order to explore the nonlinear relationship between GITwps and BW and to enhance the homoscedastic variance of residuals and the predictive ability of the models, the four linearly selected equations were refitted using the natural logarithm transformation of GITwps and the predictor variable BW. All nonlinear models (Table 3) demonstrated similar fit statistics when compared with the linear models (Appendix A). However, the nonlinear models revealed more pronounced differences between species at the intercept than the linear models. 

### 3.2. Model Evaluation

Figure 3 displays the results of standardized regression coefficients (SRC) for the developed models predicting GITwps (g/kg BW) in small ruminants fed *ad libitum*. In all equations, there was a negative relationship with BW (kg) and DMI (g/kg BW) and a positive relationship with NDFI (g/kg BW or g/g DMI), albeit with varying intensities (i.e., different SRC values). For both the linear Equations (5)–(8) and the nonlinear Equations (9)–(12), the impact of BW on GITwps prediction decreased as NDFI (g/kg BW or g/g DMI) was introduced in the model. Additionally, while Equations (7) and (11) showed a positive relationship with NDF as relative BW intake (NDFI, g/kg BW), its inclusion as diet concentration (NDFI, g/g DMI; Equations (8) and (12)) demonstrated the greater impact on the prediction of GITwps for both sheep and goats.

Figure 4 illustrates the results of the 4-fold cross-evaluation for the linear models predicting GITwps in small ruminants fed ad libitum. Equations (5)–(8) utilizing the predictor variables BW (kg) only, BW plus DMI (g/kg BW), or BW plus NDFI (g/kg BW) demonstrated similar fit statistics. However, the model incorporating NDFI (g/g DMI; Equation (8)) as a predictor variable exhibited a higher CCC (0.59) and lower RMSPE (38.9). The decomposition of the MSEP revealed that 0.015 of the error was associated with mean bias, 0.013 was attributed to systematic bias, and 0.97 was attributed to random errors. Additionally, the precision (*R*^2^) of this model was higher (0.42) compared with the models with only BW (0.37), BW plus DMI (0.38), and BW plus NDFI g/kg BW (0.2). 

The cross-evaluation of the non-linear models (Figure 5) revealed that the model incorporating NDFI (g/g DMI; Equation (12)) as a predictor variable was the best model for predicting GITwps. It demonstrated a CCC of 0.64 and an RMSPE of 38.3. The decomposition of MSEP indicated that 0.03 of the error was associated with mean bias, 0.01 was attributed to systematic bias, and 0.96 was due to random errors. Additionally, the precision (*R*^2^) of this model was higher (0.48) compared with the models utilizing only BW (0.38), BW plus DMI (0.43), and BW plus NDFI g/kg BW (0.38) for predicting GITwps in small ruminants fed ad libitum.

## 4. Discussion

According to the prediction models, the type of diet (i.e., suckling or solid diet) and feeding level (i.e., ad libitum or restricted) determine the pattern of the GITwps:BW relationship. Specifically, models have shown that the GITwps are influenced by BW only in animals feeding on a solid diet and that diet quality plays an important role in the GITwps for small ruminants fed ad libitum. Furthermore, it was determined that the GITwps:BW relationship is independent of sex (i.e., intact males, castrated males, or females); however, it is influenced by species (i.e., sheep or goats) in individuals fed ad libitum. Nevertheless, when fed ad libitum, pregnant females exhibited a notable reduction in GITwps, particularly in cases of multiple pregnancies (i.e., two or more fetuses), regardless of species. 

### 4.1. Goats Fed with the Ad Libitum Suckling Diet

The group fed a milk-based diet consisting of thirty-six goats with mean GITwps (g/kg BW) of 130 (±62) and mean BW (kg) of 5.4 (±0.55). In this specific group, no relationship between GITwps and BW was observed (Equation (1)). This result was expected due to the small variation in weight among these animals at slaughter. As their GITwps were exclusively dependent on a milk diet, it allowed for rapid transit of digesta throughout the gastrointestinal tract.

### 4.2. Animals Fed with Feeding Levels of Solid Diet

Small ruminants fed a solid diet comprised a group of 714 individuals with a mean GITwps of 201.08 (±56) g/kg BW and BW of 29.27 (±13.21) kg. The range for GITwps was from 39.8 to 376 g/kg BW, while the range for BW was from 5 to 66.8 kg. In this specific case, a strong linear negative relationship between GITwps and BW was observed (Equation (2)), applicable to both sheep and goats. This means that GITwps was proportional to BW, indicating that as the animals’ BW increased, the ratio of GIT pool size to BW decreased (Figure 1). In a study by Gindri et al. [48], a quadratic relationship was found between the total GITwps (g) and BW (kg) in Saanen goats weighing between 15 and 45 kg. 

Furthermore, in the current study, it was also observed that restriction levels only affected the intercept of the GITwps:BW relationship. Animals subjected to feed restriction (Equation (4)) exhibited a smaller GITwps of 5.13 (g/kg BW) compared with small ruminants fed ad libitum (Equation (3)). Although restricted animals likely attempted to increase mean retention time (MRT) to improve feed digestibility, the lower feed intake resulted in a smaller GIT wet pool size. The highest level of feed restriction assessed in this meta-regression was 60% of the intake of the animals fed ad libitum, meaning that the animals in this group consumed only 40% of the amount ingested by their ad libitum-fed counterparts. The mean GITwps for animals under feed restriction was 210 (g/kg BW), ranging from 46 to 360 (g/kg BW). Gut fill is closely related to feed intake and the quality of the selected diet [49]. Intake is directly proportional to maintenance requirements, which decrease per unit of BW as animals grow [50]. While the level of feed intake is generally negatively related to digesta MRT [51,52,53,54], doubling the intake leads to only a 20 to 40% decrease in particle MRT in the forest. Additionally, some researchers have not found any significant effect of the intake level on MRT [55,56].

Sex is known to have an impact on feed intake, GIT capacity, and feed digestibility [57,58,59]. In the case of small ruminants fed a solid diet, it was observed that sex influenced GITwps only at the intercept. Intact males had greater GITwps compared with castrated males and females. The quantity of feed consumed is regulated to maintain a constant intake of digestible energy [60]. Recent meta-regression studies have shown that the net requirements for growth, maintenance, and energy utilization efficiency for growth differ between the sexes of growing goats [61,62], which may influence feed intake (i.e., intact males have higher feed intake than females and castrated males; [58,59]). Therefore, differences in feed intake among sexes may explain the observed variations in the GIT:BW relationship in the current study. In addition, previous research [59] found that the sex effect on dry matter intake is constant throughout animal growth, which agrees with the absence of a sex effect on the slope found in this study. Besides that, body weight and sex are correlated variables, so it would be expected that there would be no interaction between them on GITwps prediction; it was once observed that BW reduced predictive power on GITwps as NDF intake was introduced in the model.

Furthermore, species also seem to influence the intercept and slope of the relationship between GITwps and BW (*p* = 0.022) in small ruminants fed a solid diet. Sheep exhibited a GITwps of 300.96 (±15.5, g/kg BW), while goats had a GITwps of 232 (±9.3 g/kg BW). Ruminants are often classified into concentrate selectors, intermediate feeders, and grazers based on morphological criteria and feeding habits [63]. Goats are considered intermediate feeders and have developed a feeding strategy between concentrate selectors and grazers [64,65,66]. On the other hand, domestic sheep are classified as grazers and are adapted to utilize poor-quality fibrous feed, possessing large, well-subdivided fermentation chambers [65]. Consequently, grazers are more efficient in utilizing fibrous feed compared with concentrate selectors, which may explain the observed differences in the GITwps:BW relationship between species. 

While feed shortages can occur on some farms, it is important to note that our primary focus is to assess the ability of body weight to reduce gut fill. Feed-restriction trials may not be suitable, as animals tend to eat to satisfy their energy requirements. Additionally, feed restriction may cause changes in the rumination pattern, mean retention time, and passage rate, thereby potentially misleading our understanding of gut-filling content along the gastrointestinal tract. Therefore, only animals fed ad libitum were selected to continue with the exploration of the GITwps:BW relationship.

### 4.3. Animals Fed with an Ad Libitum Solid Diet

The physiological state is another characteristic that can influence the GITwps:BW relationship. Pregnant small ruminant females fed ad libitum showed a reduction in GITwps at the intercept for single pregnancies (0.708 g/kg BW) and multiple pregnancies (i.e., two or more fetuses; 0.909 g/kg BW) after 50 days of pregnancy, compared with non-pregnant animals, regardless of species. This reduction in GITwps can be attributed to the increased size of the fetus and annexes, which compresses the rumen and other segments of the GIT, thereby decreasing GIT capacity [67]. Additionally, studies have shown that reticulorumen MRT decreases during the last third of gestation, regardless of changes in feed intake and GIT capacity [7,68]. These findings indicate that factors other than intake and GIT capacity modulate MRT during the last third of gestation.

When the variable sex was tested only for small ruminants fed ad libitum, it no longer influenced the prediction of GITwps. Similarly, Gindri et al. [48] found that the sex did not affect GIT wet pool size or GIT wet tissue in growing goats. Furthermore, sex did not impact any of the model parameters for predicting the growth of GITwps in goats [59]. In contrast, species influenced the intercept and slope of the GITwps:BW relationship in small ruminants fed ad libitum (Table 2; Equation (5)), with sheep showing a greater intercept and slope compared with goats. Tsiplakou et al. [69] compared goats and sheep fed the same diet, same food quantity, and same forage-to-concentrate ratio and found that sheep had significantly longer rumen retention time (RRT) (30.03 h vs. 14.43 h), MRT (40.80 h vs. 27.81 h), and shorter transit time (TT) (8.86 h vs. 11.49 h) compared with goats. Previous studies have also reported that the MRT of undigested residues in the GIT is longer in sheep than in goats [70,71,72,73,74]. These findings align with our results, where sheep exhibited a larger GITwps of 83.96 (g/kg BW; Table 2; Equation (5)) compared with goats at the same BW.

In the current meta-regression, studies conducted by Oliveira et al. [38] (n = 34) and Costa et al. [34] (n = 47) evaluated sheep-fed diets containing different levels of metabolizable energy, and this difference in diet quality could promote a longer MRT in the GIT of these sheep, consequently resulting in a larger GITwps compared with the other studies carried out with sheep. Another explanation for our results could be that goats have a smaller proportion of the gut in relation to BW, leading to rapid movement of digesta from the rumen and throughout the gastrointestinal tract [75]. On the other hand, for the non-linear equations, a more consistent and evident effect of species was observed, modifying the GITwps:BW relationship only at the intercept, with sheep showing higher tract content compared with goats (Table 3; Equation (9)). 

When variables related to diet components were evaluated, similar responses were observed in both the linear (Table 2) and non-linear (Table 3) prediction models for GITwps. The inclusion of DMI (g/kg BW) influenced the GITwps:BW relationship for both sheep and goats (Table 2; Equation (6)). In sheep, an increase in DMI resulted in a decrease in GITwps. This finding was consistent with the non-linear equation, which showed that an increase in DMI (g/kg BW) led to a reduction in GITwps for sheep according to e^(−0.014145×DMI (g/kg BW))^ (Table 3; Equation (10)). As previously discussed, goats and sheep have different feeding behaviors, GIT capacity, and passage rate of digesta; these species-specific differences probably promote the difference at intercept observed in Equations (6) and (10). Other studies indicated that MRT can be modulated by feed intake and GIT capacity, and whether intake increases or GIT capacity remains constant MRT, decreases [1]. Furthermore, the passage of the grass through the digestive tract increased both with the amount given and with increased comminution [76] as a result of a reduction of initial volume and retention time in the reticulorumen [77]. This difference is proportionately greater in mature animals than in young animals [18]. Therefore, the chopped roughage hay provided to animals among the studies in this meta-analysis and the great range of animals’ BW analyzed (5 to 50 kg) could explain the reason for a decreased GITwps (g/kg BW) as increasing DM intake level (g/kg BW).

The effect of NDF intake (g/kg BW) on the GITwps was independent of species, and the GIT content increased by 2.84 g/kg BW with NDFI increasing (Table 2; Equation (7)). In accordance with the non-linear equation, an increase in NDF intake (g/kg BW) promotes an increase in GITwps by e^(0.00788×NDFI (g/kg BW))^ (Table 3; Equation (11)), without differences between species, i.e., the NDF intake increases the GITwps in the same proportion in both goats and sheep. The only difference is that sheep have a higher GITwps by 82 g/kg BW than goats due to intercept differences.

On the other hand, the concentration of NDF intake in the diet (g/g DMI) removed the effect of species in the linear equation (Table 2; Equation (8)). This result was also observed in the non-linear equation (Table 3; Equation (12)), where dietary NDFI concentration did not interact with species, leading to an increase in GITwps content by e^(1.01×NDFI (g/g DMI))^ for both sheep and goats. These findings support previous studies that have highlighted the differences between goats and sheep in terms of dietary preferences, physiological characteristics, and anatomical adaptations related to diet [65].

Moreover, the inclusion of NDF in the equations significantly reduced the predictive power of BW on GITwps (Figure 3). Neutral detergent fiber and its digestibility play a crucial role in regulating forage intake in ruminants. NDF is considered the primary component that limits rumen fill and is highly correlated with rumination time and the amount of chewing required to reduce feed volume. These factors ultimately impact the filling effect on the GIT and the escape of particles from the reticulorumen. Van Soest et al. [78] established that NDF is more closely related to the daily ruminating time, GIT filling, and DMI compared with other fractions such as crude fiber and acid detergent fiber. Almeida et al. [58] also emphasized the influence of NDF in predicting intake and recommended considering dietary NDF when developing empirical equations. Our equations, which incorporate NDF content as a predictor, suggest that the NDF content in the diet (NDFI, g/g DMI) has a greater influence on GITwps than BW alone.

## 5. Conclusions

The variation of the GITwps in small ruminants with a solid-fed diet was accurately explained by the animal’s body weight; however, the impact of BW on GITwps prediction decreased as NDF intake expressed as diet concentration (g/g DMI) was included in the model and therefore should be considered in studies evaluating GIT fill. The developed models are relevant as they may help researchers understand the factors influencing GITwps in small ruminants, providing valuable insights into their digestive processes and nutritional requirements; furthermore, they may help measure weight gain and empty body weight without requiring animal slaughter or inaccurate markers.

## Figures and Tables

**Figure 1 animals-13-02909-f001:**
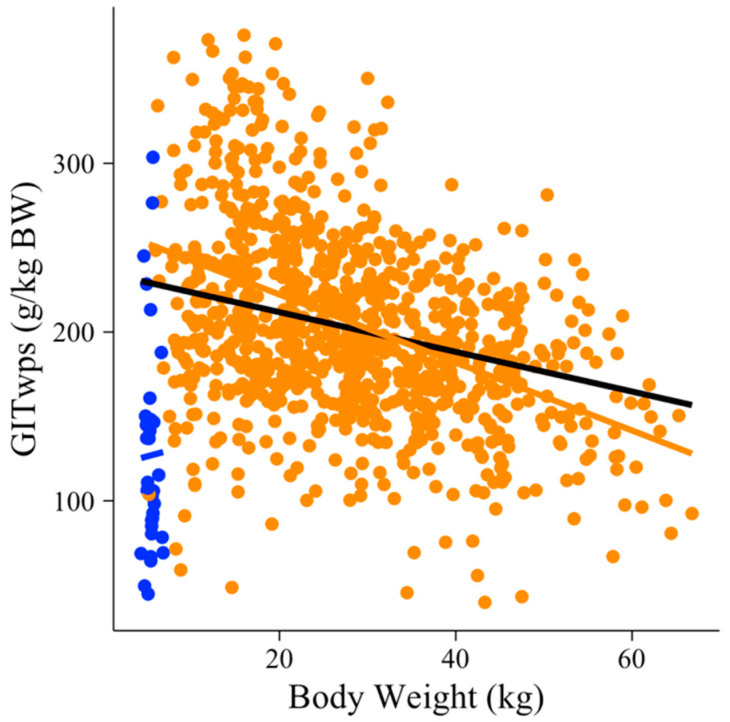
Small ruminants’ gastrointestinal wet pool size (GITwps; g/kg BW) along the range of body weight (BW; kg) of all individuals in the dataset (black line) or suckling (blue dots and line) and solid diet animals (orange dots and line).

**Figure 2 animals-13-02909-f002:**
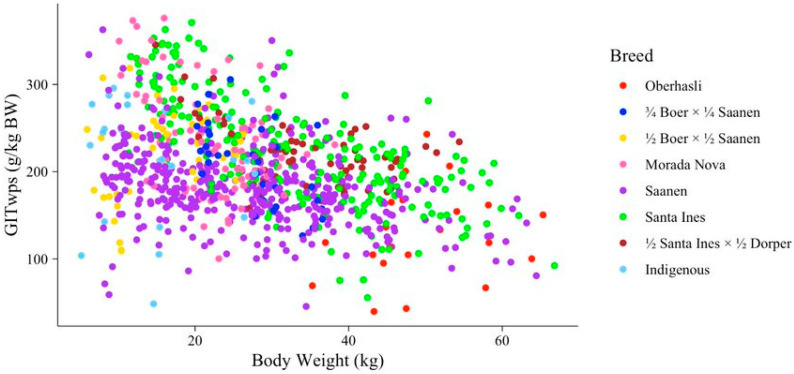
Breed distribution along gastrointestinal wet pool size (GITwps; g/kg BW) and body weight (BW; kg) relationship for small ruminants fed solid diet.

**Figure 3 animals-13-02909-f003:**
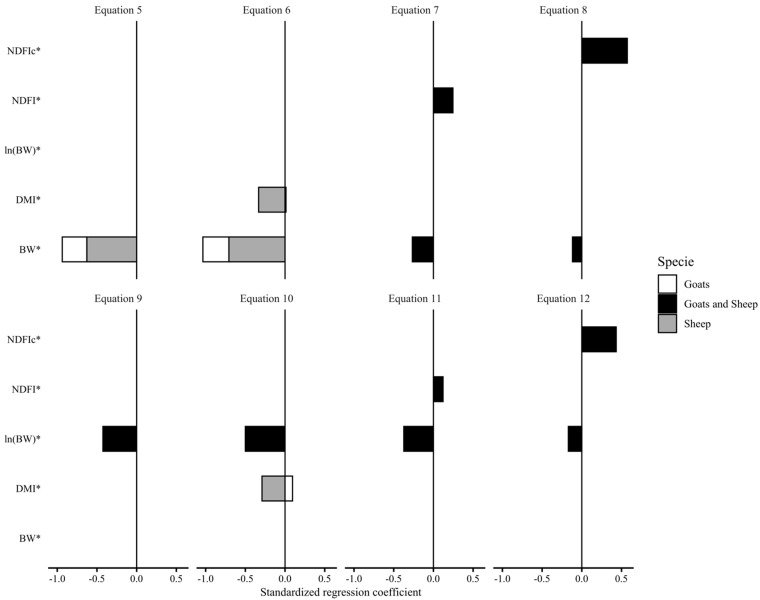
Standardized regression coefficients obtained from standardized (*) predictor and predicted variables of the developed models of gastrointestinal wet pool size (GITwps; g/kg BW) in growing small ruminants fed ad libitum (Equation (5), Goat: GITwps g/kg BW = 223.85 ± 14.38 − 1.36 ± 0.42 × BW kg, Sheep: GITwps g/kg BW = 309.22 ± 21.47 − 2.77 ± 0.58 × BW kg; Equation (6), Goat: GITwps g/kg BW = 225.04 ± 21.14 − 1.42 ± 0.42 × BW kg + 0.10 ± 0.66 × DMI g/kg BW, Sheep: GITwps g/kg BW = 399.55 ± 33.69 − 3.07 ± 0.62 × BW kg − 2.88 ± 0.84 × DMI g/kg BW; Equation (7), Goat and Sheep: GITwps g/kg BW = 200.77 ± 16.7 − 1.15 ± 0.35 × BW kg + 2.84 ± 0.66 × NDFI g/kg BW; Equation (8), Goat and Sheep: GITwps g/kg BW = 110.11 ± 17.46 − 0.5 ± 0.33 × BW kg + 238.05 ± 21.8 × NDFI g/g DMI; Equation (9), Goat: GITwps g/kg BW = 525.5 ± 1.18 × BW kg^−0.33±0.05^, Sheep: GITwps g/kg BW = 653.8 ± 1.05 × BW kg^−0.33±0.05^; Equation (10), Goat: GITwps g/kg BW = 573.9 ± 1.2 × BW kg^−0.39±0.05^ × e^0.00461±0.003×DMI g/kg BW^, Sheep: GITwps g/kg BW = 1185.8 ± 1.14 × BW kg^−0.39±0.05^ × e^−0.014145±0.005×DMI g/kg BW^; Equation (11), Goat: GITwps g/kg BW = 430.04 ± 1.23 × BW kg^−0.29±0.05^ × e^0.00788±0.004×NDFI g/kg BW^, Sheep: GITwps g/kg BW = 512.2 ± 1.06 × BW kg^−0.29±0.05^ × e^0.00788±0.004×NDFI g/kg BW^; Equation (12) Goat: GITwps g/kg BW = 185.2 ± 1.27 × BW kg^−0.13±0.06^ × e^1.01±0.14×NDFI g/g DMI^, Sheep: GITwps g/kg BW = 200.55 ± 1.08 × BW kg^−0.13±0.06^ × e^1.01±0.14×NDFI g/g DMI^).

**Figure 4 animals-13-02909-f004:**
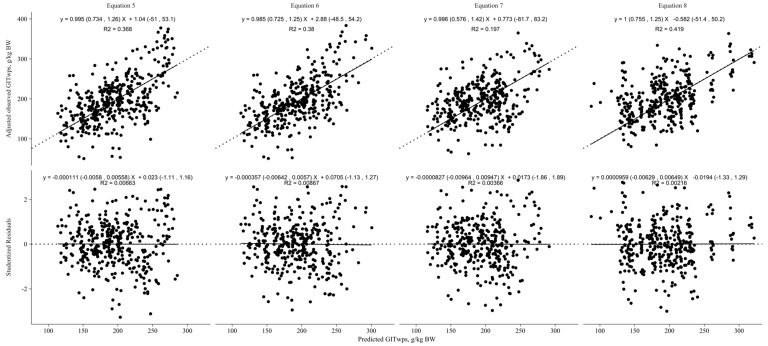
Linear models’ cross-evaluation results for predicting gastrointestinal wet pool size (GITwps; g/kg BW) in growing small ruminants fed ad libitum. Regression between adjusted observed GITwps, adjusted for the random effect of study, and predicted GITwps, in the top part of the graphics, or studentized residuals, in the bottom part of the graphics, using the developed models (Equation (5), Goat: GITwps g/kg BW = 223.85 ± 14.38 − 1.36 ± 0.42 × BW kg, Sheep: GITwps g/kg BW = 309.22 ± 21.47 − 2.77 ± 0.58 × BW kg; Equation (6), Goat: GITwps g/kg BW = 225.04 ± 21.14 − 1.42 ± 0.42 × BW kg + 0.10 ± 0.66 × DMI g/kg BW, Sheep: GITwps g/kg BW = 399.55 ± 33.69 − 3.07 ± 0.62 × BW kg − 2.88 ± 0.84 × DMI g/kg BW; Equation (7), Goat and Sheep: GITwps g/kg BW = 200.77 ± 16.7 − 1.15 ± 0.35 × BW kg + 2.84 ± 0.66 × NDFI g/kg BW; Equation (8), Goat and Sheep: GITwps g/kg BW = 110.11 ± 17.46 − 0.5 ± 0.33 × BW kg + 238.05 ± 21.8 × NDFI g/g DMI).

**Figure 5 animals-13-02909-f005:**
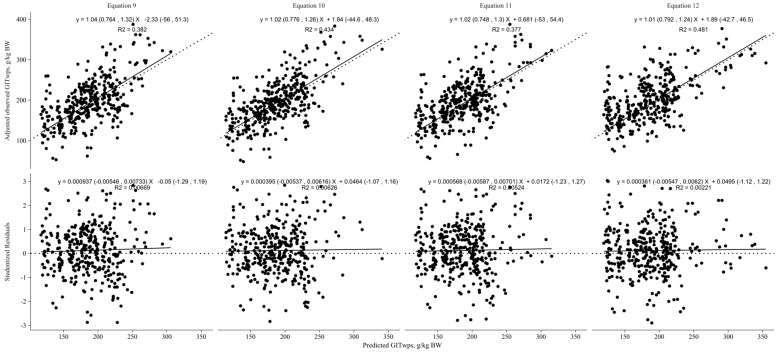
Nonlinear models’ cross-evaluation results for predicting gastrointestinal wet pool size (GITwps; g/kg BW) in growing small ruminants fed ad libitum. Regression between adjusted observed GITwps, adjusted for the random effect of study and predicted GITwps, in the top part of the graphics, or studentized residuals, in the bottom part of the graphics, using the developed models (Equation (9), Goat: GITwps g/kg BW = 525.5 ± 1.18 × BW kg^−0.33±0.05^, Sheep: GITwps g/kg BW = 653.8 ± 1.05 × BW kg^−0.33±0.05^; Equation (10), Goat: GITwps g/kg BW = 573.9 ± 1.2 × BW kg^−0.39±0.05^ × e^0.00461±0.003×DMI g/kg BW^, Sheep: GITwps g/kg BW = 1185.8 ± 1.14 × BW kg^−0.39±0.05^ × e^−0.014145±0.005×DMI g/kg BW^; Equation (11), Goat: GITwps g/kg BW = 430.04 ± 1.23 × BW kg^−0.29±0.05^ × e^0.00788±0.004×NDFI g/kg BW^, Sheep: GITwps g/kg BW = 512.2 ± 1.06 × BW kg^−0.29±0.05^ × e^0.00788±0.004×NDFI g/kg BW^; Equation (12) Goat: GITwps g/kg BW = 185.2 ± 1.27 × BW kg^−0.13±0.06^ × e^1.01±0.14×NDFI g/g DMI^, Sheep: GITwps g/kg BW = 200.55 ± 1.08 × BW kg^−0.13±0.06^ × e^1.01±0.14×NDFI g/g DMI^).

**Table 1 animals-13-02909-t001:** Summary of animal characteristics in the 21 studies used to assemble the dataset used in the meta-regression.

Study	Sex ^1^	Breed ^2^	Species	Diet ^3^	n ^4^	BW ^5^ (kg)	GITwps ^6^ (g/kg BW)	DMI ^7^ (g/kg BW)	NDFI ^8^ (g/kg BW)	NDFI (g/g DMI)
[21]	FM	S	Goat	Suckling	12	4.6–6.8	49.5–303.6	–	–	–
[22]	M	S	Goat	Suckling	8	5.1–5.7	44.7–105.9	–	–	–
[23]	M	I	Goat	Suckling	6	4.9–6.6	98.3–187.9	–	–	–
[24]	M	I	Goat	Suckling	4	4.9–5.1	144.9–146.2	–	–	–
[25]	M	BS	Goat	Suckling	6	4.3–6.7	68.7–160.8	–	–	–
[26]	CFM	S	Goat	100, 75, or 50%	55	28–47	103.7–243.8	14.9–26	4.6–7.7	0.28–0.33
[21]	CFM	S	Goat	100, 75, or 50%	46	6–16.7	163–362.8	12–26.9	3.9–8	0.31
[27]	M	BBS	Goat	100, 70, or 40%	21	21–36.6	126.6–305.6	18–34	7–13	0.37–0.41
[28]	C	S	Goat	100, 70, or 40%	27	21–35.5	100–286.6	21–41	8.9–17	0.41
[29]	CFM	S	Goat	100, 75, or 50%	54	17–34	86–239	11- 30.5	3.7–10	0.33
[30]	F	S	Goat	100, 70, or 40%	15	24.8–46	45.5–247.6	12–28	3.7–8.7	0.31
[31]	M	S	Goat	100, 70, or 40%	18	29–51	103–319.8	15–31	4.7–9.8	0.31–0.34
[32] ^9^	F	S or OB	Goat	100%	42	33–65	39.8–243	15–31	6.4–13	0.42–0.46
[22]	M	S	Goat	100, 70, or 40%	27	9.6–21.6	165.7–248.6	13–25.9	6.7–13	0.51
[23]	M	I	Goat	100%, or TM20	12	8.8–27.8	147.7–295.7	17–23.6	9–12.8	0.54
[24]	M	I	Goat	100%, or TM20	16	5–16.9	48.6–277	0.5–28	0.3–15	0.54
[33]	C	S	Goat	100%, or MT	36	9.7–31	124.9–243	14–42.8	5–16	0.38
[25]	M	BS	Goat	100, 70, or 40%	18	5.9–16	170–318.6	3.8–18.9	1.5–7	0.38
[25]	M	BS	Goat	100, 70, or 40%	21	14.5–26	190–295	22–47.8	8.6–19	0.35–0.40
[34]	M	MN	Sheep	100%	47	10–31.6	100–376	20.8–53	5.6–32	0.27–0.72
[35]	F	SID	Sheep	100, 70, or 40%	24	20–42.6	167–307	15.8–32.9	–	–
[36] ^9^	F	SI	Sheep	100%	83	29–66.8	55.7–287	15–49.8	7–30.5	0.23–0.77
[37]	M	SID	Sheep	100, 70, or 40%	30	27.6–54	174–251.7	16–32	8–15	0.46–0.50
[38]	M	SI	Sheep	100%	34	11–33.5	167.5–370.9	19.9–33	6.6–20	0.26–0.73
[39]	M	SI	Sheep	100, 70, or 40%	24	29.6–47	125.8–238.7	8.7–32.7	3.6–13.7	0.42
[40]	CM	SI	Sheep	100, 70, or 40%	29	16–33	147–344	18–31	9–15	0.49
[41]	CFM	MN	Sheep	100, 70, or 40%	35	13.7–38	137–276.8	19–32	8–14	0.44

^1^ C = castrated males; F = females; M = intact males. ^2^ S = Saanen; I = Indigenous; BS = ½ Boer × ½ Saanen; BBS = ¾ Boer × ¼ Saanen; OB = Oberhasli; MN = Morada Nova; SID = ½ Santa Ines × ½ Dorper; SI = Santa Ines. ^3^ Suckling = animals fed exclusively with milk; after weaning animals were fed a solid diet at different intake levels, such as 100% = feeding ad libitum; 40%, 50%, 70%, or 75% intake of the ad libitum; MT = maintenance level; TM20 = 1.2 × MT. ^4^ n = total number of records in the study. ^5^ BW = body weight. ^6^ GITwps = gastrointestinal tract wet pool size. ^7^ DMI = dry matter intake. ^8^ NDFI = neutral detergent fiber intake. ^9^ Studies with pregnant females.

**Table 2 animals-13-02909-t002:** Selected linear equations for predicting gastrointestinal wet pool size (GITwps) of growing small ruminants fed ad libitum.

Equations	Species	Description ^1^	n	RMSE ^2^	RMSE%	CCC ^3^
(5)	Goat	GITwps (g/kg BW) = 223.85 (±14.38) − 1.36 (±0.42) × BW (kg)	390	43.01	22.37	0.60
	Sheep	GITwps (g/kg BW) = 309.22 (±21.47) − 2.77 (±0.58) × BW (kg)				
(6)	Goat	GITwps (g/kg BW) = 225.04 (±21.14) − 1.42 (±0.42) × BW (kg) + 0.10 (±0.66) × DMI (g/kg BW)	365	41.12	21.39	0.65
	Sheep	GITwps (g/kg BW) = 399.55 (±33.69) − 3.07 (±0.62) × BW (kg) − 2.88 (±0.84) × DMI (g/kg BW)				
(7)	Goat and Sheep	GITwps (g/kg BW) = 200.77 (±16.7) − 1.15 (±0.35) × BW (kg) + 2.84 (±0.66) × NDFI (g/kg BW)	357	41.96	21.82	0.62
(8)	Goat and Sheep	GITwps (g/kg BW) = 110.11 (±17.46) − 0.5 (±0.33) × BW (kg) + 238.05 (±21.8) × NDFI (g/g DMI)	357	36.77	19.12	0.74

^1^ BW = body weight; DMI = dry matter intake; NDFI = neutral detergent fiber intake. ^2^ Root means square prediction error. ^3^ Lin’s concordance correlation coefficient.

**Table 3 animals-13-02909-t003:** Selected nonlinear equations for predicting gastrointestinal wet pool size (GITwps) of growing small ruminants fed ad libitum.

Equations	Species	Description ^1^	n	RMSE ^2^	RMSE%	CCC ^3^
(9)	Goat	GITwps (g/kg BW) = 525.5 (±1.18) × BW (kg)^−0.33 (±0.05)^	390	42.11	21.90	0.62
	Sheep	GITwps (g/kg BW) = 653.8 (±1.05) × BW (kg)^−0.33 (±0.05)^				
(10)	Goat	GITwps (g/kg BW) = 573.9 (±1.2) × BW (kg)^−0.39 (±0.05)^ × e^(0.00461 (±0.003) × DMI (g/kg BW))^	365	40.22	20.94	0.66
	Sheep	GITwps (g/kg BW) = 1185.8 (±1.14) × BW (kg)^−0.39 (±0.05)^ × e^(−0.014145 (±0.005) × DMI (g/kg BW))^				
(11)	Goat	GITwps (g/kg BW) = 430.04 (±1.23) × BW (kg)^−0.29 (±0.05)^ × e^(0.00788 (±0.004) × NDFI (g/kg BW))^	357	41.48	21.58	0.64
	Sheep	GITwps (g/kg BW) = 512.2 (±1.06) × BW (kg)^−0.29 (±0.05)^ × e^(0.00788 (±0.004) × NDFI (g/kg BW))^				
(12)	Goat	GITwps (g/kg BW) = 185.2 (±1.27) × BW (kg)^−0.13 (±0.06)^ × e^(1.01 (±0.14) × NDFI (g/g DMI))^	357	36.2	18.83	0.75
	Sheep	GITwps (g/kg BW) = 200.55 (±1.08) × BW (kg)^−0.13 (±0.06)^ × e^(1.01 (±0.14) × NDFI (g/g DMI))^				

^1^ BW = body weight; DMI = dry matter intake; NDFI = neutral detergent fiber intake. ^2^ Root means square prediction error. ^3^ Lin’s concordance correlation coefficient.

## Data Availability

The data presented in this study are available upon request to the corresponding author.

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
