# Peer review of "Modeling Gastrointestinal Tract Wet Pool Size in Small Ruminants"

_animals, 2023, doi:10.3390/ani13182909_

Round 1
Reviewer 1 Report
Respected authors
All opinion was emitted with all respect to the efforts of the authors for the preparation of the experiment and its report
The purpose of this study was to present a linear and non-linear model (using a meta-regression approach) for predicting gastrointestinal wet pool size (GITwps) of small ruminants. Due that roughly 80% of OM digestible is fermented in rumen; then, understanding the dynamics of GITwps is crucial for developing effective nutritional strategies, as well as the prediction of dressing percentage. Justification of the experiment is clear and the mathematical and statistical procedure appropriates to dully contrast the objective raised. The developed models were evaluated using a 4-fold cross-evaluation repeated over a thousand iterations. However, it would be appropriate, using the same information, to compare its accuracy with other current models.
In my personal opinion, this manuscript can be considered for publishing in the high quality journal as is Animals Journal. Only few flaws must be corrected before it considered to be publishing
Specific
L26: The predictions models confirms that..
L27: Additionally, the physiological stage, such as..
Abstract is ok, but correlation coefficient must be indicated in L39, L40, L44 and 45.
Keywords: Can improved (include more specific keywords) Hint: small ruminants; GIT fill, prediction models; diet, intrinsic factors
Introduction
L54: “animal growth” is a physiological state. Then, rewrite as: diet composition [4,5], and physiological state, such as growth and pregnancy
L57: “nutritional strategies” It's a very broad concept. maybe “feeding strategies” is more appropriate
L68: variation in BW measurements and consequently affects the precision in the prediction of weight gain, as well as, empty body weight
Material and methods
Why the general procedure to determine digesta weight was not exposed? Please, include this information.
BW it’s a very important variable in your predictions. Please define clearly if the final body weight was registered as fasted weight, or as full weight (without fasting). Please specify.
L79: Rewrite in this order here and through the document: different sexes (intact males, n=218; castrate males, n=116, and females, n=110)
L97-100: Is confusing. Feed intake level was calculated between daily intake and BW on the day slaughter? How did that happen? Did you include the data of feed consumption from the last day prior to slaughter? Moreover, NDF intake was calculated according to daily intake. The daily intake is the average daily intake registered during each trial? Please, it’s very important that specify and clarify.
L124: DM intake level
L126: Please write properly! (hint) NDFI expressed either, as daily intake (NDFI, g/d) or a BW relative intake (NDFI, g/kg BW).
L139-150: The developed models were evaluated using a 4-fold cross-evaluation repeated over a thousand iterations. However, it would be appropriate, using the same information, to compare its accuracy with other current models.
Results
L161 and 164: square r value for the 1,2,3 and 4 equations?
Discussion
L300: Please, avoid “In this study, we developed models to predict GITwps in small ruminants” is leftover. Start the sentence as: According to the prediction models, type of diet and intake level (ad libitum or restricted) determine…
L303: Avoid terms like “we found” “we observed’ for this type of study. Please rewrite (Hint): Specifically, models shown that the GITwps is influenced by BW only in animals feeding on a solid diet, and that the diet quality plays an important role in the GITwps for small ruminants fed ad libitum.
Conclusion
Make a conclusion about of the objective! Which are the factors included in the models that best explain the variation of the GITwpsin small ruminant. What applicability (certainty and scope) offer the developed models. Please focus your discussion mainly on this.
Author Response
Dear Reviewer,
Please check the responses to your comments in the attached file, thank you.
Best regards,
Authors.

Reviewer 2 Report
Dear Authors,
Thank you for the work on wet pool size in small ruminants.
This is an interesting approach for an unusual topic as rumen fill and volume is generally related to BW to the power of 1 in cattle, so I'm surprised it is not similar in sheep if not goats.
Overall, the study presents data describing rumen pool size among breeds, species, sex and physiological ranges. Essentially, you are trying to predict rumen water/fluid balance and did not address water intake and that might be part of the missing information. Please add some discussion about water intake and water balance across the rumen wall as that is likely important in your calculations. How much water was consumed in each study from feed, versus free-choice water? That might also impact your estimations.
Line 140: in estimating the predicted variable.
178-180: at that stage of growth, sex is likely a proxy for BW and why the intercept is the only difference. Please discuss. And as you point out, once you add, NDF intake, BW disappears and that is likely part of the reason.
409-412: Why the difference in wet weight in sheep vs goats and why did DMI result in less wps? That does not make intuitive sense or mechanistic sense as more DMI should result in greater weight unless there is more digestibility with greater intake because of the diets. Please spend more time discussing this and try to provide a mechanism.
448: I'm not sure they will be both accurate and precise. I would argue accurate and useful.
There are some data from Antonello Cannas that might be helpful for this evaluation and manuscript.
Author Response

(The authors gave the same response as above.)

Reviewer 3 Report
Dear authors,
Your manuscript titled ¨Modelling gastrointestinal tract wet pool size in small ruminants" is addressing an interesting topic of research for being published at Animals after minor revision has been made by the authors. A list of comments is sent to you to improve its quality before accepting it. Please, pay attention to all of them and resubmit a new article version after that.
- Use italics all over the document for mentioning the term 'ad libitum'.
- Check the separation of syllables in all the sentences where a hyphen is present respecting the correct spelling rules for each word hyphenated.
- Describe the meaning of the acronyms EBW, AICc, RRT, MRT and TT.
- Use scientific nomenclature to cite all the plant species mentioned.
- Add units to RRT, MRT and TT in L391-L392.
- Reformulate Conclusions (try to not enumerate the results got).
- Check all references cited according to Animals' instructions for authors.
Yours sincerely,
Reviewer.
Author Response

(The authors gave the same response as above.)

Round 2
Reviewer 1 Report
Dear Authors
I have read the revised manuscript and appreciate the authors' consideration of my previous suggestions. Authors have covered all of my observations in an acceptable manner, in such a way that I have no further observations.